# The microbiota of *Amblyomma americanum* reflects known westward expansion

**Luis Martinez-Villegas** [1], **Paula Lado** [2], **Hans Klompen** [3], **Selena Wang** [4], **Caleb Cummings** [4], **Risa Pesapane** [4,5], **Sarah M. Short** [1] *

**1** Department of Entomology, The Ohio State University, Columbus, Ohio, United States of America,
**2** Department of Evolution, Ecology and Organismal Biology, The Ohio State University, Columbus, Ohio, United States of America, **3** Department of Evolution, Ecology, and Organismal Biology and Museum of Biological Diversity, The Ohio State University, Columbus, Ohio, United States of America, **4** Veterinary Preventive Medicine, College of Veterinary Medicine, The Ohio State University, Columbus, Ohio, United States of America, **5** School of Environment and Natural Resources, College of Food, Agricultural, and Environmental Sciences, The Ohio State University, Columbus, Ohio, United States of America

☯ These authors contributed equally to this work.
¤ Current address: Paula Lado, National Bio and Agro-Defense Facility, USDA Agricultural Research Service (ARS), Manhattan, Kansas, United States of America
* short.343@osu.edu

**Data Availability Statement:** Nucleotide sequences were deposited in the Sequence Read Archive and are accessible with the accession PRJNA976750.

## Abstract

*Amblyomma americanum*, a known vector of multiple tick-borne pathogens, has expanded its geographic distribution across the United States in the past decades. Tick microbiomes may play a role shaping their host's life history and vectorial capacity. Bacterial communities associated with *A. americanum* may reflect, or enable, geographic expansion and studying the microbiota will improve understanding of tick-borne disease ecology. We examined the microbiota structure of 189 adult ticks collected in four regions encompassing their historical and current geographic distribution. Both geographic region of origin and sex were significant predictors of alpha diversity. As in other tick models, within-sample diversity was low and uneven given the presence of dominant endosymbionts. Beta diversity analyses revealed that bacterial profiles of ticks of both sexes collected in the West were significantly different from those of the Historic range. Biomarkers were identified for all regions except the historical range. In addition, Bray-Curtis dissimilarities overall increased with distance between sites. Relative quantification of ecological processes showed that, for females and males, respectively, drift and dispersal limitation were the primary drivers of community assembly. Collectively, our findings highlight how microbiota structural variance discriminates the western-expanded populations of *A. americanum* ticks from the Historical range. Spatial autocorrelation, and particularly the detection of non-selective ecological processes, are indicative of geographic isolation. We also found that prevalence of *Ehrlichia chaffeensis*, *E. ewingii*, and *Anaplasma phagocytophilum* ranged from 3.40–5.11% and did not significantly differ by region. *Rickettsia rickettsii* was absent from our samples. Our conclusions demonstrate the value of synergistic analysis of biogeographic and microbial ecology data in investigating range expansion in *A. americanum* and potentially other tick vectors as well.

**Funding:** LM and SMS were supported by the Ohio State University Infectious Diseases Institute (https://idi.osu.edu/) and the Ohio State University College of Food, Agricultural, and Environmental Sciences (https://cfaes.osu.edu/). RP, SW, and CC were supported by the Ohio State University College of Food, Agricultural, and Environmental Sciences (https://cfaes.osu.edu/) and the Ohio State University College of Veterinary Medicine (https://vet.osu.edu). the funders had no role in study design, data collection and analysis, decision to publish, or preparation of the manuscript.

**Competing interests:** The authors have declared that no competing interests exist.

## Introduction

Prevalence of multiple tick-borne diseases (TBDs) e.g., Lyme disease, anaplasmosis, ehrlichiosis, and spotted fever group rickettsiosis have substantially increased in the past 12 years in the United States [1]. The pathogens that cause these diseases are transmitted by hard ticks primarily in the genera *Ixodes* (Lyme, anaplasmosis), *Amblyomma* (ehrlichiosis, rickettsiosis), and *Dermacentor* (rickettsiosis). The reasons behind these increases are multifactorial, but one main cause is hypothesized to be the geographic range expansion of the ticks that transmit these pathogens [2, 3]. Among them, *Amblyomma americanum*, historically found in the southeastern USA [4], has rapidly expanded its geographic range in the past decades reaching Michigan, Pennsylvania, and New York in the north, while extending to Nebraska and South Dakota in the west [5–7]. In addition, reports of its presence in Canadian provinces include Alberta, British Columbia, Manitoba, Quebec and Ontario [8]. In many vector models [9–12], climate and anthropogenic environmental changes are critical interacting drivers of tick range expansion, e.g., by making habitats more amenable, or altering host density and distribution [2, 13–17]. In *A. americanum*, shifts in land use and host population densities may be the critical drivers of expansion. Additionally, it is hypothesized that *A. americanum's* recent expansion is at least in part a re-occupation of its ancestral range [18–20]. Understanding how the biology of ticks varies along environmental gradients and landscapes may prove critical to improve our TBD control efforts [21–25]. One relevant aspect of tick biology is the tick microbiota, i.e., the bacterial community associated with ticks throughout their life cycle.

The advancement of high throughput sequencing (HTS) has led to a more thorough profiling and characterization of the microbial communities associated with ticks [26]. Although some functional roles of the tick microbiota have been elucidated, the field is far from fully explaining the intricacies of host-microbiota interactions in ticks [26–28]. Nonetheless, evidence of bacterial community structuring based on habitat or geographic region of origin (henceforth, region) has been detected in several tick hosts, predominantly *Ixodes* and *Dermacentor* [29–35]. This suggests the microbiota may have relevance for understanding tick range expansions and has led to calls for more microbiome studies spanning more tick species in wider geographic ranges [36]. By broadening the spatial scales of tick microbiota studies, we may improve our understanding of the role of the microbiota in TBD ecology.

In addition to its potential role in informing the occurrence of tick range expansion, the tick microbiota has other effects that may facilitate range expansion. For instance, bacterial pathogens such as *Anaplasma* and *Borrelia* induce physiological changes impacting their *Ixodes* hosts' survival, questing, mobility, and feeding activity (reviewed in [37]). The increased synthesis of heat shock proteins [38] or antifreeze glycoproteins [39] induced by *Anaplasma* infection could improve tick survival when moving over long distances. *Borrelia* infected *Ixodes* nymphs have higher fat content, which may confer a benefit when colonizing challenging habitats (cold weather and/or high elevation) [40]. Beyond these effects, the tick microbiota also plays fundamental roles in tick biology. As obligate blood feeders, ticks rely on endosymbionts (*Rickettsia*, *Coxiella*-like, *Francisella*-like, *Candidatus Midchloria*) to obtain vitamin B and cofactors that are either scarce or not readily available in blood [41–43]. Experimental and computational evidence also suggests a role of commensal and symbiont bacteria in tick host reproductive fitness [44], immune response [45–47]; and vector competence [48, 49]. Predictive functional studies of tick microbiota are few [50, 51] but, in combination with taxonomic profiling and functional genomics, could enhance our understanding of the microbiome's influence on host biology [50, 52]. Thus, and as in other vector models [49, 53, 54], the study of the tripartite vector-pathogen-microbiome interactions is critical to understanding vector biology and ecology.

We recently attempted to detect signal of *A. americanum* range expansion at the genetic level by analyzing the population structure of the same ticks used for this study [18]. Lado et al. [18] results suggested that ticks from the edge populations, which possessed low diversity, may be at the front of expansion based on genetic theory [55, 56]. Therefore, we sought to determine whether microbiota structure could provide an alternative ecological signal of range expansion in these same individuals. A recent study in *Ixodes ricinus* revealed that bacterial community structures reflected range expansion along elevation clines, supporting use of this approach for our question [57]. Here, we analyzed microbiota spatial structuring between expanded and historic range populations of female and male *A. americanum* and the extent to which geographic distance and ecological forces drive bacterial community turnover. Furthermore, we tested for the presence of known vertebrate bacterial pathogens to identify signature microbiota profiles predictive of tick infection status.

## Materials and methods

### Sampling of tick specimens

The adult unfed ticks used in this study are the same as Lado et al. [18]. We caught all ticks in the wild by dragging a cloth through the ground vegetation and then preserving them in 95% ethanol. When needed, authorities of the Department of Natural Resources from OH, MI, and IN were contacted in advance to collect samples. Otherwise, samples were collected in locations where a park ranger could grant permission on-site, or in public access locations. Some samples were shared by collaborators as stated in the acknowledgments. In total the dataset is comprised of 189 *A. americanum* specimens (females = 98; males = 91) collected at 24 localities in 17 states distributed across the historic and expanded ranges for this species according to Monzón et al. [7] (see Table 1, and S1 File for metadata). As in [18], informed by available literature and consensus [7], we divided the samples into 4 regions to which we refer throughout the manuscript as: (i) Historic (HC), corresponding to the historic range for this species; (ii) Midwest (MW), located within the expanded range and containing the samples from OH; (iii) West (W), encompassing samples from the expanded range collected from KS and OK; and (iv) Northeast (NE), encompassing samples from the expanded range collected from RI, MD, and VA. The rationale behind this experimental design was to contrast the microbiota of specimens from populations that have been long established to those from populations considered to be at edge zones along the expanded range [18]. Therefore, diversity analyses presented in this study represent comparisons between the above-mentioned regions, and not individual States, as that would create more bias than what can be already expected from field collections. Detailed information for all ticks can be found at the Ohio State Acarology Collection (OSAL) database accessible at: (https://acarology.osu.edu/database).

**Table 1. Summarized sample metadata[1].**

| Region | States included in region | Sample size by sex |
|---|---|---|
| HC | 12 | 68 females, 61 males |
| NE | 3 | 14 females, 13 males |
| MW | 1 | 5 females, 7 males |
| W | 2 | 11 females, 10 males |

[1]. Full metadata for all samples can be found in S1 Table in S1 File.

## DNA extraction

The DNA samples employed here are the same as in our population structure study [18]. Briefly, we extracted genomic DNA from each individual tick following methods reported previously [35]. All the ticks, in random order, were surface sterilized twice using commercial bleach (Clorox, NaOCl active bleach) at 3% for one minute, rinsed with distilled water and then washed with 95% ethanol for one minute. We then extracted DNA from each individual using the Qiagen Dneasy Blood and Tissue kit as per manufacturers' instructions with one modification as in [58]: a cut was made at the posterior-lateral part of the tick idiosoma to allow ATL buffer to reach all tissues. Afterwards we recovered the cuticle (exoskeleton) of each tick to use as voucher specimens. Their corresponding OSAL accession numbers are listed in Table 1. Genomic DNA was then quantified using a Qubit 3.0 fluorimeter, aliquoted and stored at -80˚C for subsequent downstream high-throughput sequencing and pathogen PCR screening.

## 16S rRNA library preparation and high throughput sequencing (HTS)

Genomic DNA (gDNA) from each of the 189 specimens was normalized to 10ng/µl and then, along with two negative controls, shipped to Argonne National Laboratory for library preparation and HTS following in-house standard operations as a single batch. We included two negative controls representing extraction and library blanks as suggested by [59] aiming to detect potential spurious environmental taxa. For taxonomic profiling through HTS we targeted the V4-V5 hypervariable region (product size of 411 bp) with the 515F-Y / 926R primers [60, 61] in a 2 X 250 bp paired-end run with an Illumina MiSeq Next Generation sequencing system. This primer set generates diversity profiles concordant with those produced with V4 amplicons [60], while enhancing phylogenetic resolution [61]. Our amplicon choice falls within the predominant targets in other tick microbiome studies (reviewed by [26]), which mostly sequenced one or more targets within the V1-V4 regions. Approximately 200 to 500 bp reads provide quality taxonomic resolution to identify many, though not all, bacterial species [61]. Therefore, we determined that sequencing the V4 region (~254 bp in length), which is most informative among the 9 hypervariable regions for detecting diversity within tick microbiomes [62], plus the V5 region ensured the most thorough bacterial community survey. This primer set has been successfully used in other microbiome studies [63–66], including ticks [35] with similar library preparation parameters [35, 63–65]. Demultiplexed raw sequences were provided by the sequencing facility.

## Bioinformatic analyses

R code for all the procedures described below can be found in S2 File.

 **(i) Amplicon sequence variant (ASV) detection, taxonomic assignment, decontamination, and additional processing.** Raw fastq sequences were analyzed using *DADA2* v1.16 [67] in R v4.2.1 [68] within the Rstudio environment v2022.7.1.554 [69] following the workflow and using the default parameters suggested by the developer at: (https://benjjneb.github.io/dada2/tutorial.html). After executing the quality filtering, trimming, and sample inference steps we detected that the forward and reverse sequences were not merging optimally. We attempted to rectify this using different truncation lengths for the forward and reverse reads and allowing for shorter minimum overlaps as in [65, 66], but this did not satisfactorily recover our merging output. Bioinformatic approaches have been proposed to overcome the problem of non-merging reads, but these approaches have not been thoroughly evaluated for ASVs [70]. Therefore, we considered that the best option was to carry out the downstream workflow with quality filtered forward reads only as suggested by the *DADA2* developer (https://github.

com/benjjneb/dada2/issues/761). As the core denoising algorithm is applied before merging to increase accuracy, nearly all substitution errors have been removed before this step [61]. The use of single end reads does result in sequences that are shorter in length than expected. In this scenario, the extent to which identities can accurately be resolved at low taxonomic levels is hindered, but not the validity of the ASVs and their derived profiles and diversity estimations [70]. It is known that use of OTUs can influence diversity estimates (inflation), taxonomic resolution, and reproducibility [71, 72]. However, use of single versus paired-end reads does not change the relationships between bacterial community profiles observed with Bray-Curtis distances [73, 74]. Other microbiome studies targeting arthropod vectors [75, 76], and human samples [77] have successfully utilized this processing approach as well.

After creating the ASV count table and removing chimeras, we proceeded to assign taxonomy using the *DADA2* formatted RDP training set (version 18). We then imported both the ASV count and taxonomy tables to the *phyloseq* R package v1.30.0 [78] for downstream microbial ecology analyses. We then removed ASVs not identified as k_Bacteria and those classified as mitochondria or chloroplasts from the *phyloseq* object. With the information provided by the blank samples we used the *decontam* R package v1.6 [79] with its default prevalence filtering parameters to remove ASVs identified as contamination. Both control samples were then removed from the dataset. Finally, as in [35], we reduced the dimension of the dataset with an abundance and prevalence filtering step. We removed ASVs accounting for fewer than 10 total reads and represented in fewer than 10% of the samples to reduce the chances of sequencing errors being identified as rare ASVs that could impact the diversity analyses. This was particularly relevant given our choice to work with forward reads only.

**(ii) Alpha diversity.** Measures of alpha diversity including observed taxonomic richness, Chao-1, ACE, Shannon, and Simpson indices were calculated for each sample with *phyloseq* (S3 File). We chose to focus our analysis and diversity plots primarily on the Gini-Simpsons' diversity index (1-D) [80] as it allows for a clear and intuitive ecological interpretation, while also being less affected by sequencing coverage [81] and spurious elements in the count tables [82]. We evaluated effects of region and sex on alpha diversity using a GLM with a quasibinomial distribution followed by a Tukey's test. Boxplots were performed with the *ggpubr* v0.4.0 [83] and GLM was run using *stats* v4.1.3.

**(iii) Beta diversity.** First, we generated a stacked barplot with *ggplot2* v3.3.6 [84] to observe the relative abundances of the Order level taxa present within each sample. We then assessed if there were differences in bacterial community structures between ticks based on Bray-Curtis (BC) dissimilarities estimated on Hellinger-transformed ASV abundances generated with the *decostand* function in the *vegan* v2.6.2 R package [85]. We chose to use BC dissimilarities and Hellinger-transformed data because our dataset contained many zeros. In datasets such as these (ones with joint absences or double zeroes), one approach is to standardize the data (as reviewed in [86]). We chose to use the Hellinger transformation because it reduces the weight of rare taxa (ASVs with low counts and many zeroes) when estimating community structure differences [87], thereby resulting in a conservative approach. Furthermore, BC dissimilarities are suited for this type of community data analysis as they exclude the effect of joint absences [88]. In addition to this, the Hellinger transformation solves the problem of the linear assumptions in PCoA ordinations [89]. After transforming the data and calculating BC dissimilarities, we performed a PCoA with *vegan* for each tick sex. We chose to use PCoA because it is compatible with dissimilarities, such as BC [90]. We used a permutational multivariate analysis of variance (PERMANOVA) [91] to test the effects of region and sex on beta diversity using the *adonis2* function in the R package *vegan*. We also split the dataset by sex and ran pairwise PERMANOVA with the *pairwiseAdonis* v0.4 R package [92] to test differences between each region.

We also tested for ASVs that significantly discriminated between tick region and sex. For this, we applied indicator species analysis [93], which relies on community ecology principles to identify ASVs that significantly reflect their niche states (p < 0.05). We ran the "multipatt" function within the *indicspecies* package v1.7.0 using the "Indval.g" function to address uneven groups. Plots were created to report indicator ASVs above arbitrary Indicator statistic (Indic-stat) thresholds. When considering tick sex, markers with an Indicstat >0.80 for males and the only female marker were plotted. When considering region, markers with and Indicstat > 0.60 were plotted.

**(iv) Mantel correlogram.**    To study spatial autocorrelation, we separated the dataset by sex and used Mantel correlograms to evaluate how bacterial composition signal behaved across a range of geographic distances given the coordinates at which samples were collected. We generated the Mantel correlograms with the function "mantel.correlog" in *vegan* performing 999 randomizations and progressive FDR corrections to test for significance. Mantel correlo-grams, like semivariograms, show the level of autocorrelation in samples from different geo-graphic locations (Table 1). They also reveal whether distance in one measure correlates positively or inversely with increasing distance in the other index. To generate a Mantel corre-logram, we estimated spearman correlations between a matrix of geographic distances and corresponding Bray-Curtis distances over a spatial scale arbitrarily assigned using Sturges' for-mula [94]. We interpreted the correlograms as suggested by [95].

**(v) Estimation of ecological processes driving community turnover.**    We used null-model analysis to evaluate the contribution of ecological processes on the bacterial community assembly. All the required steps were performed using the *trans_nullmodel class* function from the *microeco* R package v0.9.0 [96] following the tutorial workflow provided at: (https:// chiliubio.github.io/microeco_tutorial/model-based-class.html#trans_nullmodel-class). This analysis is based on the methodology described by [97, 98] based on Vellend's conceptual framework [99]. We used this method to quantify the relative influence of ecological processes shaping within-tick bacterial structures. The sequence of analyses includes the calculation of the phylogenetic signal, beta mean nearest taxon distance (βMNTD), beta nearest taxon index (βNTI) and Bray-Curtis-based Raup-Crick (RC$_{Bray}$) measures. Briefly, we first obtained the βMNTD from all the pairwise comparisons among ticks from the same sex. The measured dif-ference between observed βMNTD and the null model, is termed βNTI, which we then used in combination with measured RC$_{Bray}$ dissimilarities as classifying parameters to assign how the calculated proportions pertain to deterministic or stochastic processes following the interpre-tation criteria proposed by [97]. These criteria can be summarized as follows: β-NTI scores greater than + 2 indicated variable selection pressures whereas scores less than -2 indicated homogeneous selective pressures. Based on the RC$_{Bray}$ index, we then inferred the contribution of the stochastic processes as follows: the fraction of pairwise comparisons with RC$_{Bray}$ between -0.95 and +0.95 pertained to drift acting alone. The fraction with RC$_{Bray}$ < -0.95 indi-cated homogeneous dispersal, and finally the fraction with RC$_{Bray}$ > +0.95 was assigned to dis-persal limitation. We tabulated the fractions assigned to each process grouping them by tick sex and visualized the comparison with a bubble plot using *ggplot2*.

## Tick-borne pathogen (TBP) detection using qPCR

Probe-based real-time PCR (qPCR) singleplex assays using primers and probes with estab-lished sensitivity and specificity were used to detect the vertebrate pathogens *Ehrlichia chaf-feensis*, *E. ewingii*, *Rickettsia rickettsii*, and *Anaplasma phagocytophilum* [100–102]. A 20μL reaction containing Taqman Fast Advanced Master Mix (Applied Biosystems, Waltham, MA) and 2μL of tick gDNA template was performed with thermocycling conditions of 95˚C for 3

minutes then 50 cycles of 95˚C for 3 seconds followed by 60˚C for 45 seconds. In each reaction, a no-template control of molecular-grade nuclease-free water and a synthetic gBlock fragment (Integrated DNA Technologies) of *E. chaffeensis* (AF403711.1), *E. ewingii* (AY428950.1), *R.* (A1G_04230), or *A. phagocytophilum* (AY151054.1) were included as controls. Samples were considered positive if a characteristic amplification curve was observed below a cycle threshold of 40. All positives were run in duplicate. Refer to S4 File for a summary table of the PCR primers.

## Results

### Sequencing overview

We sequenced the bacterial community of 189 *A. americanum* ticks and two negative controls. This resulted in 6,609,083 high-quality reads. After decontamination and removal of the control samples, our data set was comprised of 6,017,532 with a median number of reads per sample of 33879 (average 32328.46; range 1528–53310). After filtering the ASVs by total abundance and prevalence in the dataset, a total of 2224 ASVs remained and were used in downstream microbial ecology analyses.

### The bacterial community structure of *A. americanum* is dominated by a few taxa

When agglomerated at the Order taxonomic level (Fig 1), the most abundant ASVs within most samples were assigned to the Orders Coxiellales, Rickettsiales, and Diplorickettsiales which accounted for 82.53%, 12.27% and 1.77% of the dataset respectively, and encompassed the 9 core ASVs present in at least 90% of the samples. The top 3 ASVs mapped to *Coxiella*,

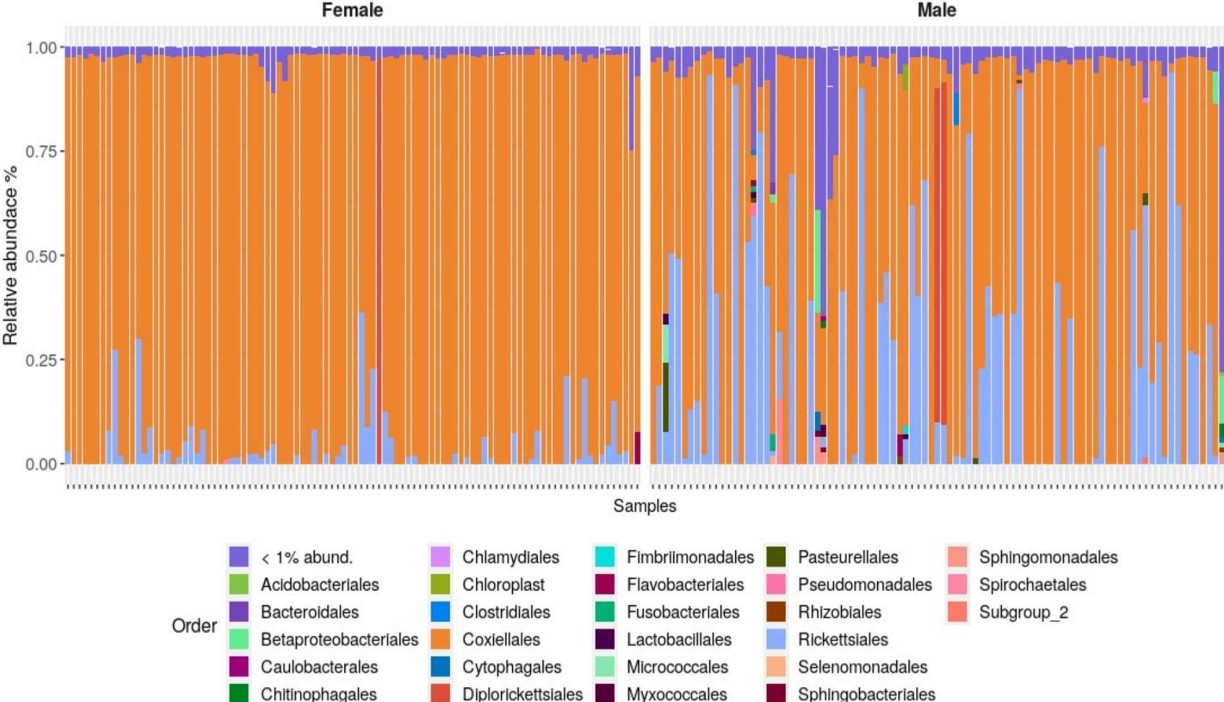

**Fig 1. Bacterial abundance profiles of individual ticks parsed by sex.** Relative abundances of the most common bacteria at the Order taxonomic level. Parsing individuals by sex revealed how community structures are dominated by few members and differ between male and female ticks.

*Rickettsia amblyommatis*, and *Rickettsiella grylli*. The first two taxa have been reported previously as common and abundant endosymbionts of *A. americanum* [44, 103–105] while the third was dominant in only 3 tick samples (Fig 1, Order Diplorickettsiales). ASVs within the Coxiellales Order were clearly dominant in females, whereas male ticks' communities were dominated by both Coxiellales and Rickettsiales with some individuals presenting more diverse profiles (Fig 1).

## Region and sex are both significant drivers of alpha diversity *A. americanum*

We estimated several alpha diversity indices for each sample (S3 File) but given the dominance of relatively few community members in most samples (Fig 1), we chose to focus on the Gini-Simpson index. Communities were mostly low in diversity and both region and sex were significant, non-interacting, predictors of Gini-Simpson indices (ANOVA following GLM p = 0.002 for region; p = 2.2e-16 for sex). NE ticks harbored significantly more diverse communities when compared to those from the HC and MW (Tukey's HSD p values of 0.01 and 0.006 respectively) but were no different than samples from W (Fig 2A). Communities associated with male ticks were more diverse than those in females (ANOVA p <0.005) (Fig 3B).

## Community structures associated with *A. americanum* ticks collected in the west significantly differ from those collected in the historic range

PERMANOVA analysis based on Bray-Curtis dissimilarities revealed that dissimilarities between tick samples can be partially attributed to the significant (and non-interacting) effect of both region and sex (Table 2). As in the case of alpha diversity, community structures in *A. americanum* and other tick models may reflect physiological and life history differences between males and females. Therefore, we split the data set by sex and explored if the clustering profiles reflected the known range expansion of *A. americanum*. As observed in both

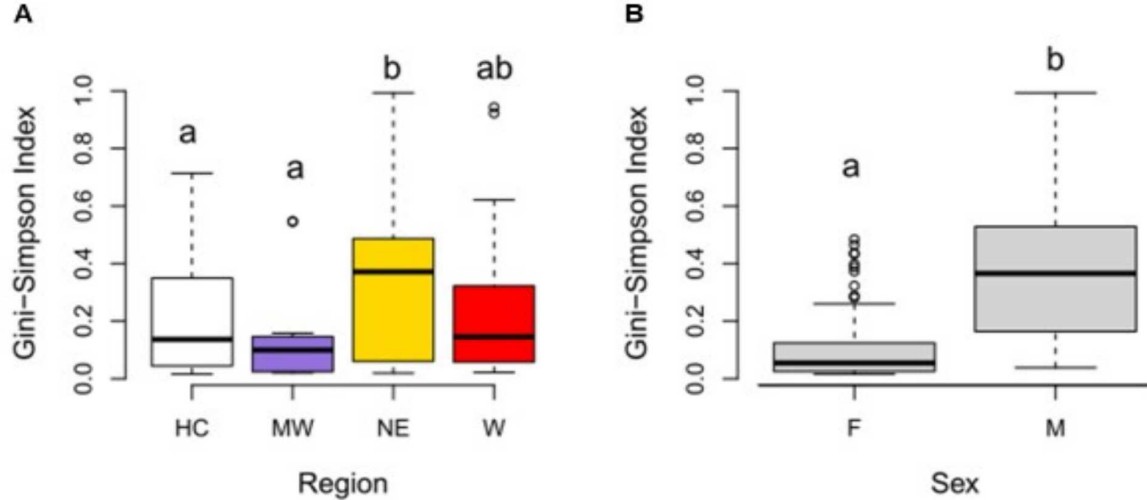

**Fig 2. Alpha diversity in bacterial communities associated with *A. americanum* significantly differ between region and sex.** Within-sample diversity, measured as the Gini-Simpson diversity index, was higher in *A. americanum* ticks from the NE (Northeast) region when compared to HC (Historic range) and MW (Midwest) ticks (A) (Tukey's HSD $p_{Nevs.HC}$ = 0.01 and $p_{Nevs.MW}$ = 0.006). Between sex comparison (B) revealed male-associated communities were significantly more diverse than female communities (ANOVA p<0.05). Sample sizes for each region are: Historic range (HC; n = 129); Northeast (NE; n = 27); Midwest (MW; n = 12); West (W; n = 21). Sample sizes for each sex are: females (F; n = 98); males (M; n = 91). Mean values are depicted by the thick back lines for each boxplot.

A

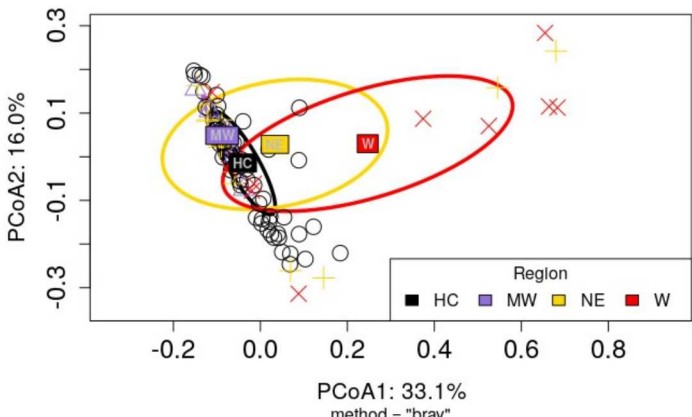

B

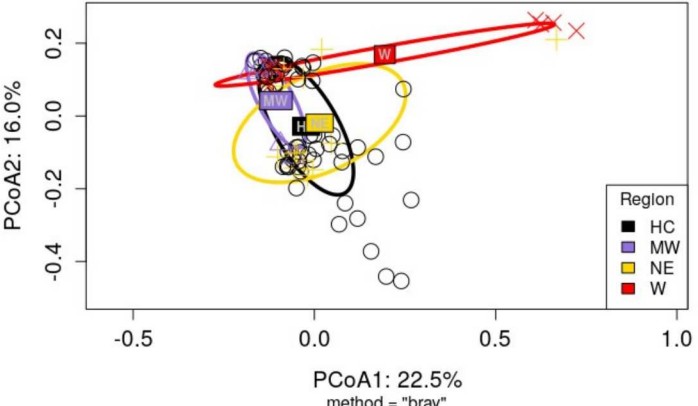

**Fig 3. Bacterial community structures of ticks from the west region are significantly different from historic range in both male and female ticks.** Ordination analysis (PCoA) based on Bray-Curtis dissimilarities among female (A) and male (B) ticks show how ticks collected in the West clustered separately from the other 3 regions predominantly along the horizontal axis (PCoA1) where the largest variances where represented (33.1% for females; 22.5 for males). Centroids for each geographic region and standard deviation ellipses follow the color code in the legend, with abbreviations representing regions as follows: HC = Historic range; MW = Midwest; NE = Northeast; and W = West. PERMANOVA analysis showed that tick sex (p = 1 x $10^{-5}$ $R^2$ = 0.2018) and region (p = 1 x $10^{-5}$ $R^2$ = 0.0569) significantly drive beta diversity (Table 2). Pairwise PERMANOVA tests (Table 3) revealed that for both tick sexes, W was significantly different from HC (p = 0.001, $R^2$ = 0.1107 for females; p = 0.001, $R^2$ = 0.0737 for males). W was also significantly different from the other two regions in both sexes except for the W vs NE comparison in females (p = 0.1470, $R^2$ = 0.0698 as shown in Table 3).

females (Fig 3A) and males (Fig 3B), W ticks had a community structure that positioned them along the PCoA dimension1 separately from the other 3 regions which closely co-localize predominantly to the left of the zero axis. A pairwise PERMANOVA (Table 3) showed that ticks of both sexes collected at W were significantly different from the ones collected at HC (p = 0.001, $R^2$ = 0.1107 for females; p = 0.001, $R^2$ = 0.0737 for males). In addition, bacterial

**Table 2. PERMANOVA assessing the association between beta diversity (quantified by Bray-Curtis dissimilarities) and explanatory variables representing region and sex.**

| Explanatory variable | $R^2$ | p—value |
|---|---|---|
| Region | 0.0569 | <1e-05 |
| Sex | 0.2018 | <1e-05 |
| Residuals | 0.7411 | |
| Total | 1.0000 | |

community structures of western ticks (females and males) were significantly different from all other regions except for W vs NE in females. Pairwise comparisons between HC, NE, and MW were non-significant (Table 3).

## Multiple indicator taxa (ASVs) of bacterial communities associated with tick sex or region

Both region and sex significantly drove dissimilarities between bacterial community structures. When comparing the regions of origin of the ticks (Fig 4A), 38 biomarkers were identified. From them, 36 were associated with ticks from W, and only one biomarker was identified for each of the MW and NE regions. No indicator ASVs were identified for Historic range samples. The predominant Phylum among the indicators was p__Protebacteria. In ticks from W, indicator ASVs were assigned to p__Proteobacteria (n = 27), p__Bacteroidetes (n = 7), p__Acidobacteria (n = 1), and p__Verrucomicrobiota (n = 1). Taxonomically, these ASVs represent 11 bacterial families (Fig 4A). In the case of MW and NE ticks, their only biomarkers were identified as p__Proteobacteria, o__Rickettsiales and p__Proteobacteria, f__Rickettsiaceae respectively (Fig 4A). Most marker ASVs were of low relative abundance (< 1%) except NE marker ASV2, which had a relative abundance of 12.23%. The ASV with the highest indicstat (0.66, p = 0.001) was an indicator for the West and was identified as f__Sphingomonadaceae. When comparing between tick sex (Fig 4B), we found 224 indicator species associated with male ticks and only one associated with females. For males, we plotted the first 24 with an indicator statistic (indicstat) > 0.80 and for females we plotted the sole indicator ASV identified (indicstat = 0.30). From the 25 reported biomarkers in Fig 4B, 23 of those associated with male ticks were assigned to p__Proteobacteria and one identified as p__Actinobacteria. The only indicator ASV reported for female ticks was identified as p__Acidobacteria. All the by-sex biomarkers had a low relative abundance in the dataset (relative abundances < 0.2%).

**Table 3. Pairwise PERMANOVA assessing the significance of the Bray-Curtis dissimilarities between ticks of the same sex when comparing W against the other regions of origin.**

| Compared regions | $R^2$ | p—value |
|---|---|---|
| **Females** | | |
| W vs HC | 0.1107 | 0.001 |
| W vs MW | 0.1318 | 0.044 |
| W vs NE | 0.0698 | 0.147 |
| **Males** | | |
| W vs HC | 0.0737 | 0.001 |
| W vs MW | 0.1639 | 0.029 |
| W vs NE | 0.1238 | 0.020 |

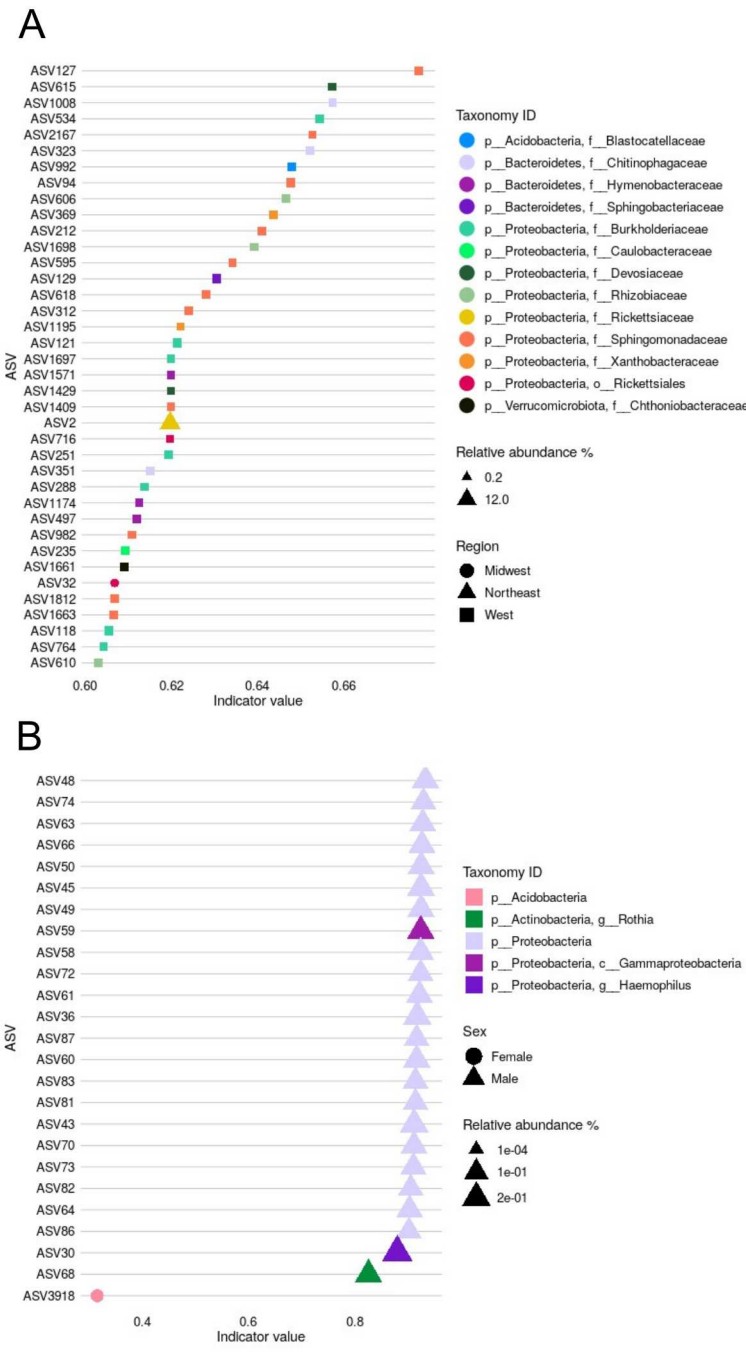

**Fig 4. Multiple taxa are significantly associated with the western region and male ticks.** Indicator analyses identified ASVs which were significantly associated with region (A), or sex (B). 36 region-specific biomarkers were associated with ticks collected in the W region, and one biomarker was identified for each of the NE and MW regions. 24 sex-specific biomarkers were associated with male ticks, whereas only one was associated with females. The relative abundance of each indicator ASV is shown by marker size, and their respective taxonomic affiliation is indicated by color code in each figure legend.

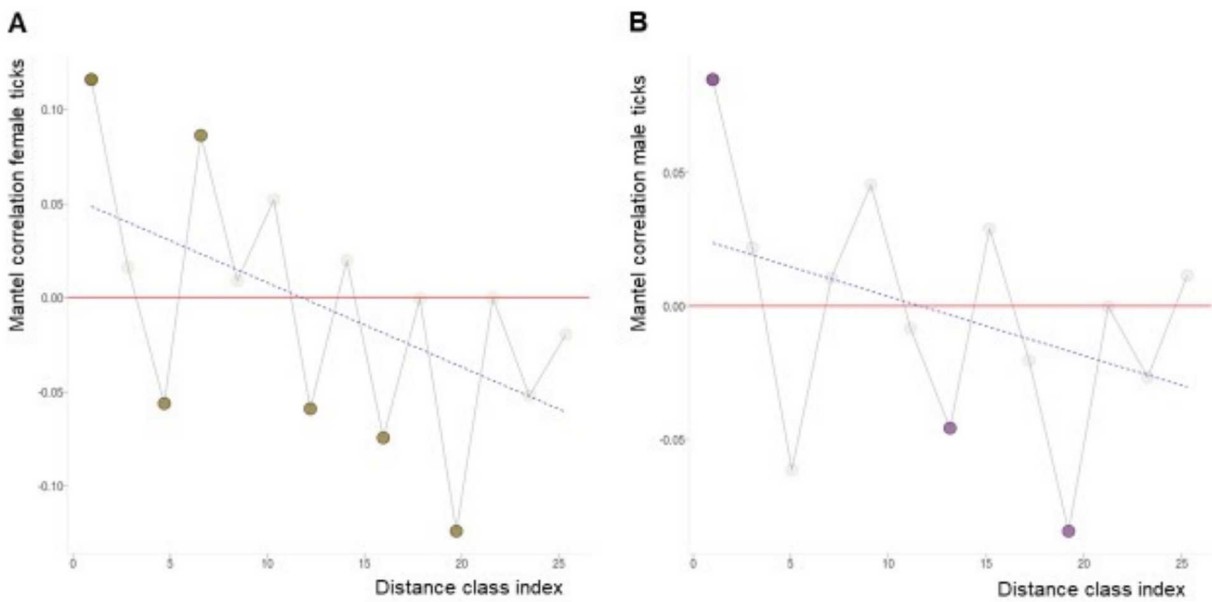

**Fig 5. Microbial communities become significantly more dissimilar with increasing geographic distance.** Mantel correlograms show relationships between Bray-Curtis dissimilarities and geographic distances among females (A) and males (B). Distance classes along the x axis are bins following the Sturges equation. Dark symbols represent significant Mantel statistics after progressive Bonferroni corrections (p < 0.05). Positive and negative correlation values indicate positive and negative relationships between Bray-Curtis values and geographic distances, respectively. Linear trendlines were added for visual purposes to highlight the nature of the correlations as distance classes increased.

### Significant spatial autocorrelation reveals the effect of geographic distance on community dissimilarities

We used a Mantel correlogram to assess autocorrelation between Bray-Curtis pairwise dissimilarities and geospatial distances among individual ticks parsed by sex (Fig 5A females; 5B males). At low distances values, the correlogram showed a significantly positive Mantel correlation (p < 0.005 after progressive Bonferroni corrections) for both sexes, indicating that ticks collected in proximal sites had similar bacterial communities (i.e., lower Bray-Curtis dissimilarity values) (opaque circles, Fig 5A and 5B). As distance increased, we qualitatively observed a "bumpy" or irregular trend towards negative correlations. We estimated more significant negative than positive correlations for both sexes at higher distance values, suggesting that the farther away the ticks were spatially, the more dissimilar their community structures measured. However, this pattern was more obvious in females, where 4 out of 6 negative correlations are significant (Fig 5A) than in males, where only 2 out of 6 negative correlations are significant (Fig 5B). The alternating pattern of positive and negative correlations for both sexes also suggests that environmental effects (or gradients) other than geographical distance may be influencing the community spatial structuring.

### Deterministic and stochastic ecological processes influence bacterial community assemblies

We quantified the relative effect of ecological processes (niche-driven or stochastic), structuring the bacterial communities associated with female and male ticks (Fig 6). Null model and phylogenetic turnover analyses revealed that for both sexes, non-selective processes (drift and dispersal) had a more prevalent role. Drift accounted for 88.11% and 85.90% of the detected effects, respectively. For both sexes, dispersal limitation was the second largest component making up

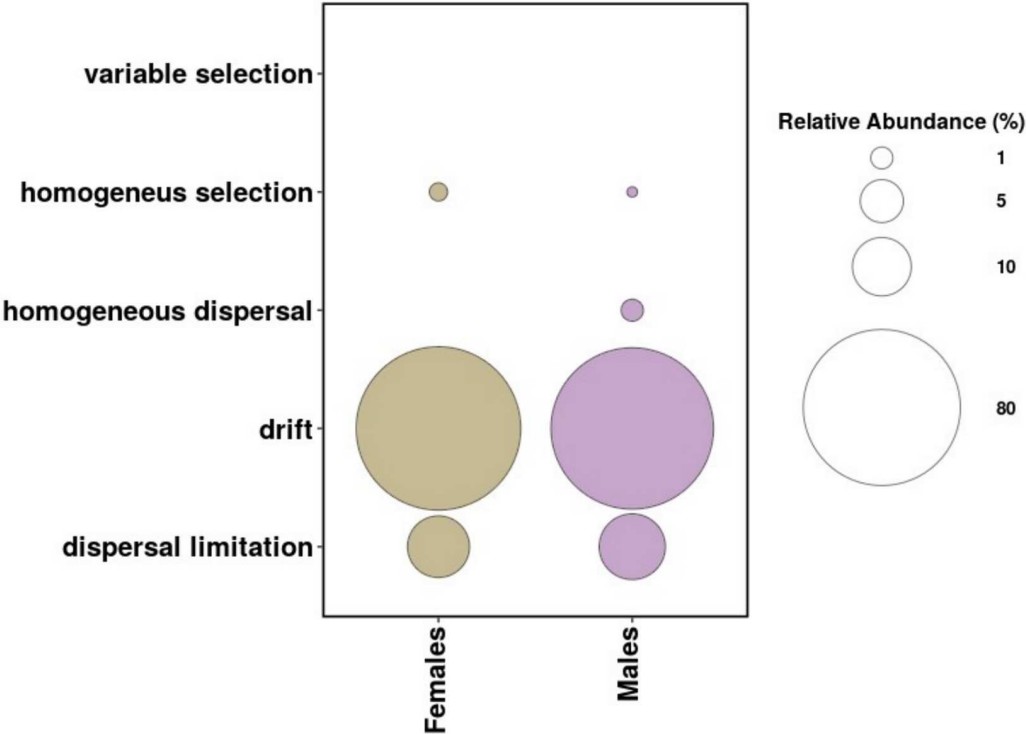

**Fig 6. Stochasticity coupled with dispersal limitation are the main drivers of microbiota dissimilarities among individuals within sexes.** Relative contribution of assembling processes structuring bacterial communities in female (beige) and male (violet) ticks. Dot sizes indicate the percentage of community turnover assigned to each type of process in agreement with the criteria presented by [97] to interpret the values of β-NTI and $RC_{Bray}$. In both female and male ticks, community turnover is predominately influenced by drift coupled with dispersal limitation.

11.23% in females and 12.96% in males. Homogeneous dispersal was detected only in male ticks, making up 1.03% of the forces shaping community structure. In sum, the largest proportions were assigned to stochastic processes which accounted for 99.34% in females and 99.91% in males. Deterministic processes were only detected in low values for both sexes, being attributed to homogeneous selection (0.66% in females and 0.09% in males). These low values suggest that niche-driven processes have a limited effect on shaping the composition of male and female associated microbiota in the geographic and temporal scales that our dataset represents.

## Molecular detection of tick-borne pathogens (TBPs)

We performed molecular diagnoses of *E. chaffeensis*, *E. ewingii*, *R. rickettsii* and *A. phagocytophilum* on 235 samples (S5 File). 158 of these samples were also included in the microbiota study. No samples from the Midwest were tested. In the complete dataset we found that prevalence of *E. chaffeensis*, *E. ewingii*, and *A. phagocytophilum* ranged from 3.40–5.11% and we failed to detect *R. rickettsii* in any sample (Table 4). When split by region, we detected *A. phagocytophilum* only in the Historic region (Table 4). When considering their locality of origin, we highlight the detection of a *A. phagocytophilum* cluster as from the 15 samples collected at Noxubee County, Mississippi, 9 were found positive for this pathogen. Moreso, one of the samples was co-infected with *E. chaffeensis* and *A. phagocytophilum*. One of the initial goals of this study was to test if bacterial community structures could predict the presence of one or multiple TBPs in *A. americanum* as attempted in other tick models with similar datasets [30].

**Table 4. Pathogen percent prevalence determined by qPCR analyses.**

|  | HC<br>n = 150 | MW<br>n = 0* | NE<br>n = 43 | W<br>n = 42 | Global<br>n = 235 |
|---|---|---|---|---|---|
| *E. chaffeensis* | 4.67 | N/A | 6.98 | 4.76 | 5.11 |
| *E. ewingii* | 2.67 | N/A | 4.65 | 4.76 | 3.40 |
| *R. rickettsii* | 0.00 | N/A | 0.00 | 0.00 | 0.00 |
| *A. phagocytophilum* | 7.33 | N/A | 0.00 | 0.00 | 4.68 |

*All genomic material was consumed during HTS

However, given the low number of infected individuals, machine learning methods could not be executed reliably, and we were unable to perform this analysis. Though not our original goal, we executed a GLM analysis to evaluate if the pathogen presence was driven by region. No significant effects were detected when making by-region pairwise comparisons for each pathogen (S6 File).

## Discussion

Our previous study suggested that *A. americanum*'s range expansion may be too recent to produce significant phylogenetic evidence, although lower levels of genetic diversity were overall observed along the edges of the distributional ranges [18]. In that work, we also found supporting evidence for a re-colonization scenario of the NE area, rather than an expansion as in the other regions. In the present study, we took an integrative 'omics approach [106] to further explore those demographic processes in *A. americanum*, with a focus on the microbiota. For both adult female and male ticks, our bacterial community analyses revealed significant spatial structuring driven by distance, particularly at the western front of expansion. The community composition snapshots also enabled us to quantify ecological processes suggestive of bacterial communities associated with hosts under geographic isolation.

Consistent with previous studies, we found that *A. americanum* harbors a bacterial community dominated by the endosymbiotic bacteria *Coxiella* [44, 103, 107–111]. Our samples generally had uneven community profiles, i.e., low richness with the majority of counts from only a few members. This diversity trend agrees with previous findings [104, 108–111] though variable numbers of observed operational taxonomic units (OTUs) are reported. Overall richness variability may be due to differences in the surface sterilization protocols which could impact measured diversity [112].

Inter-regional comparisons of the within-sample (alpha) diversity showed that Northeast ticks harbored slightly more diverse bacterial communities than individuals from the Historic range and Midwest, but not from the West. Thus, the richest communities are present in a current expansion area (W) and a region that was re-colonized by populations of this species (NE) [18]. Region has been found to be a significant driver of alpha diversity in other ticks, including *I. ricinus*, *D. variabilis*, and *D. reticulatus* [57, 113, 114]. Duncan et al. [114] found *D. variabilis* individuals from the Northeast USA to have a higher richness than those from the Midwest, South, and West populations. In their study, ticks from the West had the lowest richness and were geographically isolated from the other populations. Interestingly, in *D. reticulatus* ticks, females but not males showed regional differences in alpha diversity [113]. Thus, it is important to assess the relationship between region and sex when studying tick microbiota spatial dynamics. We showed that female ticks had significantly lower alpha diversity than males, as has been previously reported in multiple tick species including *A. americanum* [30,

57, 105, 109, 110, 114]. It is likely that the high relative abundance of *Coxiella* in females compared to males is driving this finding in our study. *Coxiella* is transovarially transmitted [115] in *A. americanum*. It is possible this inheritance mechanism underlies the dominance of *Coxiella* in female-associated microbial communities, but this requires further study [116].

Spatial structure, i.e., microbial community membership and abundance profiles influenced by geographic origin, has been reported in several ixodid ticks encompassing both short and wide ranges [29, 31–35, 117, 118]. Likewise, sex has also been reported as a significant driver of beta diversity [30, 35, 51, 113]. We found significant spatial structuring in our samples, and this was similar for both male and female ticks. Western ticks significantly differed from the other regions, except for the female W vs NE comparison. A similar study of *D. variabilis* in the U.S.A found male and female adult western ticks had significantly different community structures from ticks collected in the Northeast, Midwest, and South [114]. In *D. reticulatus*, beta diversity variance was driven by geographic origin but only in females [113]. This is in contrast to our study where we detected significant spatial structuring for both sexes.

We also identified differentially abundant and prevalent bacterial taxa driving tick microbiota community structures. We detected non-o__Rickettsiales p__Proteobacteria as predominant biomarkers of western ticks, whereas the single markers for the northeastern and midwestern ticks were ASVs identified as o__Rickettsiales. Whether the association of o__Rickettsiales ASVs with the North and Midwest populations is relevant to understanding *A. americanum*'s TBD ecology is unknown, though our qPCR analysis reveals with a high degree of certainty that these ASVs are not *R. rickettsii*. Nevertheless, given the recent reports of predicted negative and positive interactions between pathogenic taxa and the host microbiota [51, 119], we believe further studies are warranted. There was a clear imbalance in the number of biomarkers among regions, and we speculate two potential causes of this phenomenon: environmental similarities and the time scale of the range expansion. Ticks moving westward could have encountered a wider set of environmental and climatic variables, versus perhaps more homogeneous landscapes among the other three regions. As mentioned above, soil and habitat act as determinants of bacterial diversity [26], including in *A. americanum* [31, 120]. As ticks spend most of their life in the leaf litter off their host, this environment would significantly structure their microbiota. Host species can also influence tick microbial community composition [121, 122]. Host usage by ticks moving West might differ from that of the Historic range resulting in new bacterial members or alteration of their microbial community structure. It is also possible that the imbalance in biomarkers is an artifact of temporally dynamic community assemblage [36]. The extent to which the differentially abundant taxa we identified are long-term symbionts of the western population or merely transient associations in this moving population is a question that requires more elaborate experiments.

The indicator taxa analysis also revealed a clear overrepresentation of taxa associated with male ticks. A skewed ratio of biomarkers discriminating male from female *A. americanum* was also reported by Trout Fryxell and De Bruyn [31] who, similarly, reported Proteobacteria and Actinobacteria ASVs as male markers. Physiological, metabolic, and life history differences between sexes may underlie this observation. For instance, male ticks are more likely to take smaller blood meals from several different host types and are more likely to have a broader home range compared to females. Therefore, males may develop stable associations with a wider variety of microorganisms than their female counterparts [57, 114]. We hypothesize that both sex and region biomarkers are reflecting tick movement through space but may also enable it. For instance, bacteria have been associated with migration in birds [123, 124] and temperature tolerant strains of fungi can facilitate range expansion in *Atta texana* ants [125]. In ticks, bacterial members of the microbiome have been shown to influence locomotion [126] generally increasing tick mobility [37]. Whether western ticks in our study incorporated

movement-beneficial symbionts, or these bacterial taxa merely depict transient associations is a question that requires more elaborate experiments.

Correlograms for both sexes revealed a negative trend, suggesting that bacterial profiles are more different between ticks as they become more geographically distant from each other. This is consistent with previous findings in *Ixodes* ticks collected along the east coast of the U. S.A. [30]. However, we also note alternating ("bumpy") patterns in both male and female correlograms. Mostly negative but also positive, significant correlations, suggest that profile dissimilarities are not only influenced by physical distances but could also be driven by other factors such as topographical features (mountain ranges, cliffs, basins) and vegetation gradients [87, 95]. For instance, similarity in topology and climate would make it possible for physically distant ticks to possess similar microbiota when in resembling environmental conditions, and vice versa. A similar diversity to distance relationship exists in microbial communities from North and South pole water samples; given their similar habitat conditions, physical distance had no effect on community turnover [127].

We further quantified ecological processes shaping bacterial community assembly [57, 97]. We found that both deterministic and stochastic forces shaped the microbiota turnover in male and female *A. americanum* ticks but that low levels of dispersal limitation (constrained exchange of bacteria among local host communities) in combination with drift were the predominant forces. This, in accordance with Vellend's framework [99, 128], implies that dispersal limitation enables drift (or weak selection) to cause turnover in community structures [97, 129]. The coupling of these two forces is congruent with bacterial communities associated with geographically isolated hosts. Given our results, this could be predominantly occurring in ticks from the West, but our analysis did not specifically evaluate this question. More balanced sample sizes at the expansion edges would strengthen conclusions inferred from ecological process quantification when evaluating range expansion [57].

Lastly, we performed pathogen detection in the ticks. Our intent was to determine if bacterial community composition could be predictive of pathogen presence, but the pathogen prevalence in our samples proved insufficient to execute supervised learning analysis. It is worth noting that Trout Fryxell and De Bruyn [31] reported no significant effect of *Ehrlichia* or *Anaplasma* presence (PCR detected) on *A. americanum* microbial community structures. In our dataset, the overall prevalence of *E. chaffeensis* and *E. ewingii* were similar to those previously reported [3, 130]. Future studies to evaluate predictive relationships between pathogens and microbiota structure may require larger sample sizes or targeted sampling in areas of high prevalence to ensure sufficient data.

In our previous study investigating genetic signatures of range expansion in *A. americanum*, we did not detect evidence of geographic isolation [18]. Using the same samples, we investigated their associated microbial communities. We detected significant bacterial community turnover in the western expansion front potentially driven by physical distance and/or environmental differences, which recapitulated the known pattern of recent expansion and suggested potential geographic isolation, at least for the western front. Ecological processes shaping the community assemblage further support the value of microbial ecology as a relevant data layer for vector biogeographical studies. Given that the tick-microbiome-pathogen triad is far from being understood, further studies are warranted to inform control strategies of TBDs as vectors expand their range accompanied and potentially enabled by their bacterial symbionts.

## Supporting information

**S1 File. Table S1: Sample metadata.**
(TXT)

**S2 File. R script: Tick_microbiota.**
(R)

**S3 File. Table S3: Alpha diversity indices.**
(TXT)

**S4 File. Real time PCR primers and probes for pathogen detection.**
(DOCX)

**S5 File. Table S5: Pathogen detection.**
(CSV)

**S6 File. R output: GLM_effect of region on infection status.**
(TXT)

## Acknowledgments

We are grateful to Jerome Goddard, Howard Ginsberg, Andrew Hoffman, Benedict Pagac, Graham Hickling, Solny Adalsteinsson, Bruce Noden, Brent Newman, Jim Occi, Shelby Ford, Renn Tumlison, and Lance Durden who provided some of the tick specimens used in this study. We also thank The National Ecological Observatory Network, a program sponsored by the National Science Foundation and operated under cooperative agreement by Battelle Memorial Institute, for providing some specimens used in this work. In addition, we thank Lorenza Beati, curator of the USNTC for the tick records.

## Author Contributions

**Conceptualization:** Luis Martinez-Villegas, Paula Lado, Hans Klompen, Risa Pesapane, Sarah M. Short.

**Data curation:** Luis Martinez-Villegas, Risa Pesapane.

**Formal analysis:** Luis Martinez-Villegas, Paula Lado, Selena Wang, Caleb Cummings, Risa Pesapane, Sarah M. Short.

**Funding acquisition:** Hans Klompen, Risa Pesapane, Sarah M. Short.

**Investigation:** Luis Martinez-Villegas, Paula Lado.

**Methodology:** Luis Martinez-Villegas, Paula Lado, Hans Klompen, Risa Pesapane.

**Project administration:** Hans Klompen, Risa Pesapane.

**Resources:** Hans Klompen, Risa Pesapane, Sarah M. Short.

**Supervision:** Hans Klompen.

**Visualization:** Luis Martinez-Villegas, Paula Lado, Sarah M. Short.

**Writing – original draft:** Luis Martinez-Villegas, Paula Lado, Sarah M. Short.

**Writing – review & editing:** Luis Martinez-Villegas, Paula Lado, Sarah M. Short.

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
