## [Decision Letter · Decision Letter 0]

19 Jan 2024

PONE-D-23-34044The microbiota of *Amblyomma americanum* reflects known westward expansionPLOS ONE

Dear Dr. Short,

Thank you for submitting your manuscript to PLOS ONE. After careful consideration, we feel that it has merit but does not fully meet PLOS ONE’s publication criteria as it currently stands. Therefore, we invite you to submit a revised version of the manuscript that addresses the points raised during the review process.

We look forward to receiving your revised manuscript.

Kind regards,

Mai abuowarda

Academic Editor

PLOS ONE

“LM and SMS were supported by the Ohio State University Infectious Diseases Institute (https://idi.osu.edu/) and the Ohio State University College of Food, Agricultural, and Environmental Sciences (https://cfaes.osu.edu/). RP, SW, and CC were supported by the Ohio State University College of Food, Agricultural, and Environmental Sciences (https://cfaes.osu.edu/) and the Ohio State University College of Veterinary Medicine (https://vet.osu.edu).”

Reviewers' comments:

Reviewer's Responses to Questions

**Comments to the Author**

1. Is the manuscript technically sound, and do the data support the conclusions?

Reviewer #1: Yes

Reviewer #2: Yes

2. Has the statistical analysis been performed appropriately and rigorously? 

Reviewer #1: Yes

Reviewer #2: Yes

3. Have the authors made all data underlying the findings in their manuscript fully available?

Reviewer #1: Yes

Reviewer #2: Yes

4. Is the manuscript presented in an intelligible fashion and written in standard English?

Reviewer #1: Yes

Reviewer #2: Yes

5. Review Comments to the Author

Reviewer #1: The manuscript details original research on the spatial structuring of microbiota in female and male Amblyoma populations across expanded and historic ranges. The manuscript effectively uses proper English, addressing relevant questions and objectives to provide answers. I have pointed out two minor corrections that are highlighted in the text.

Reviewer #2: Overall, the manucript titled “The microbiota of Amblyomma americanum reflects known westward expansion” is an interesting study describing microbiota of Amblyomma americanum in some region of united states.

The manuscript should be revised by the authors in order to be published. I addressed some point below line by line.

The manuscript is definitely too long, thus I suggest shortening it by cutting some parts on the too detailed paragraphs (best of speech is the short and informative one (.

Title

The title is ambiguous and needs more clarification. What does the word (westward expansion) refer to?

Abstract

The abstract must involve the tick borne pathogens that determined by researchers

Introduction

Line 55: it is preferred to replace “driver” with -----motivation

Line 56: geographic range expansion---to-----geographic distribution

Line 73: You could have mentioned briefly the functional roles of the tick microbiota

Materials and Methods

Line 117: 189 A. americanum specimens collected at 24 localities, this number is too small. You can describe your limitation and difficulties to obtain more samples.

Line 148: Mention the active ingredient of bleach in parentheses.

Line 146: you are mentioned “we extracted genomic DNA from each individual tick” , This contradicts your statement “Ticks, randomly chosen, were surface”, so you must replace the second sentence with “all ticks sterilized with ……

Line 152: ATL----- Write the full name and abbreviation in parentheses

Line 157-176; line 293-303: I suggest making small table including gene, Primer type, Primer sequence, PCR product size (base pair), thermocycling conditions, References for each primer

Line 177: Bioinformatic analyses--- Authors should rephrase this section so that it is clear and concise

Results

Line 323, 324: must be written in discussion

Fig 2: it is preferred to be changed with good clear colors and resolution, so that the readers can recognize the details of the figure easily.

Fig. 3: the meaning of the abbreviated letters must be written on the figure as fig.4

Line 364: please write abbreviation in parentheses as example (W- region) or any other format to avoid the confusion with other words ---within whole manuscript

Discussion

Line 541; structuring—to ---structure

Line 587, 587; reformulate the sentence

6. PLOS authors have the option to publish the peer review history of their article (what does this mean?). If published, this will include your full peer review and any attached files.

Reviewer #1: **Yes: **Idika kalu Idika

Reviewer #2: No

---

## [Author Response · Author response to Decision Letter 0]

11 Apr 2024

Response to reviewers has been uploaded in the attached documents section.

---

## [Decision Letter · Decision Letter 1]

22 May 2024

The microbiota of *Amblyomma americanum* reflects known westward expansion

PONE-D-23-34044R1

Dear Dr. Short,

We’re pleased to inform you that your manuscript has been judged scientifically suitable for publication and will be formally accepted for publication once it meets all outstanding technical requirements.

Kind regards,

Mai abuowarda

Academic Editor

PLOS ONE

Additional Editor Comments (optional):

Reviewers' comments:

Reviewer's Responses to Questions

**Comments to the Author**

1. If the authors have adequately addressed your comments raised in a previous round of review and you feel that this manuscript is now acceptable for publication, you may indicate that here to bypass the “Comments to the Author” section, enter your conflict of interest statement in the “Confidential to Editor” section, and submit your "Accept" recommendation.

Reviewer #1: All comments have been addressed

Reviewer #2: All comments have been addressed

2. Is the manuscript technically sound, and do the data support the conclusions?

Reviewer #1: Yes

Reviewer #2: Yes

3. Has the statistical analysis been performed appropriately and rigorously? 

Reviewer #1: Yes

Reviewer #2: Yes

4. Have the authors made all data underlying the findings in their manuscript fully available?

Reviewer #1: Yes

Reviewer #2: Yes

5. Is the manuscript presented in an intelligible fashion and written in standard English?

Reviewer #1: Yes

Reviewer #2: Yes

6. Review Comments to the Author

Reviewer #1: The manuscript details original research on the spatial structuring of microbiota in female and male Amblyoma populations across expanded and historic ranges. The manuscript effectively uses proper English, addressing relevant questions and objectives to provide answers. I believe the manuscript is in good shape to be accepted for publication

Reviewer #2: All reviewer comments are addressed in revised manuscript xxxxxxxxxxxxxxxxxxxxxxxxxxxxxxxxxxxxxxxxxxxxxxxxxxxxxxxxxxxxxxxxxxxxxxxxxxxxxxxxxxxxxxxxxxxxxxxxxxxxxxxxxxxxxxxxxxxxxxxxxxxxxxxxxxxxxxxxxxxxxxxxxxxxxxxxxxxxxxxxxxxxxxxxxxxxxxxxxxxxxxxxxxxxxxxxxxxxxxxxxxxxxxxxxxxxxxxxxxxxxxxxxxxxxxxxxxxxxxxxxxxxxxxxxxxxxxxxxxxxxxxxxxxxxxxxxxxxxxxxxxxxxxxxxxxxxxxxxxxxxxxxxxxxxxxxxxxxxxxxxxxxxxxxxxxxxxxxxxxxxxxxxxxxxxxxxxxxxxxxxxxxxxxxxxxxxxxxxxxxxxxxxxxxxxxxxxxxxxxxxxxxxxxxxxxxxxxxxxxxxxxxxxxxxxxxxxxxxxxxxxxxxxxxxxxxxxxxxxxxxxxxxxxxxxxxxxxxx

7. PLOS authors have the option to publish the peer review history of their article (what does this mean?). If published, this will include your full peer review and any attached files.

Reviewer #1: **Yes: **Idika Ik

Reviewer #2: No

---

## [Editor Report · Acceptance letter]

30 May 2024

PONE-D-23-34044R1 

PLOS ONE

Dear Dr. Short, 

I'm pleased to inform you that your manuscript has been deemed suitable for publication in PLOS ONE. Congratulations! Your manuscript is now being handed over to our production team.

Kind regards, 

on behalf of

dr Mai abuowarda 

Academic Editor

PLOS ONE